

# Effects of diversity and coalescence of species assemblages on ecosystem function at the margins of an environmental shift

Jo A. Werba[1], Alexandra L. Stucy[2], Ariane L. Peralta[2] and Michael W. McCoy[2]

[1] Department of Biology, McMaster University, Hamilton, Ontario, Canada
[2] Department of Biology, East Carolina University, Greenville, NC, United States of America

## ABSTRACT

Sea level rise is mixing formerly isolated freshwater communities with saltwater communities. The structure of these new aquatic communities is jointly controlled by pre- and post-colonization processes. Similarly, since salinity is a strong abiotic determinant of post-colonization survival in coastal systems, changes in salinity will likely impact community composition. In this study, we examine how a strong abiotic gradient affects the diversity and structure of bacterial and zooplankton communities and associated ecosystem functions (decomposition and carbon mineralization). We ran a six week dispersal experiment using mesocosm ponds with four distinct salinity profiles (0, 5, 9, and 13 psu). We find that salinity is the primary driver of both bacterial and zooplankton community composition. We find evidence that as bacterial richness increases so does the amount of decomposition. A phenomenological model suggests carbon mineralization may decrease at mid-salinities; this warrants future work into possible mechanisms for this apparent loss of function. Understanding how salinization changes community structure and ecosystem function may be paramount for managing and conserving coastal plain ecosystems where salinity is increasing due to sea level rise, saltwater intrusion, storm surges, and drought.

# INTRODUCTION

Salinity is an abiotic filter for almost all aquatic organisms, and therefore strongly influences their distribution and abundance. Changes in salinity can alter the distribution of organisms (*Hall & Burns, 2002*), community assembly processes (*Jones & McMahon, 2009*), and associated ecosystem functions (*Schäfer et al., 2012*; *Więski et al., 2010*). Thus, understanding how communities are altered following changes in habitat quality is critical for predicting the consequences of environmental change.

Changes in salinity due to climate change associated sea level rise (SLR), coastal storm surges, ditching and dredging, over-extraction of aquifers, and increased input of salts from upstream sources greatly affect coastal wetlands (*Nicholls & Cazenave, 2010*; *Craft et al., 2009*). Specifically, SLR and ocean over-wash from storm surges change the chemical

Corresponding author
Jo A. Werba, jo.werba@gmail.com

make up of coastal freshwater bodies and increase the movement of organisms between salt and freshwater habitat types, creating new species assemblages by merging communities that were historically allopatric. Furthermore, increases in salinity, alkalinity, pH, and ion concentrations from salt water incursions into freshwater habitats is toxic to many freshwater organisms (e.g. *Albecker & McCoy, 2017*; *Hintz & Relyea, 2017*), creating a physiological barrier that affects the composition of freshwater communities. Changes in abiotic conditions, disturbance regime, and dispersal dynamics in coastal ponds are therefore likely affect both the composition of species and the ecological functions of the system, which can ultimately jeopardize important socio-economic services provided by these ecosystems (*De Groot, Wilson & Boumans, 2002*; *Kirwan & Megonigal, 2013*). For instance, zooplankton abundance and diversity is known to be negatively correlated with salinity (*Nielsen et al., 2008*; *Helenius et al., 2017*; *Schallenberg, Hall & Burns, 2003*), and decreased diversity is often associated with reductions in grazing rates (*Zervoudaki, Nielsen & Carstensen, 2009*), nutrient cycling (*Makarewicz & Likens, 1979*) and other downstream functions such as carbon export (*Isla, Scharek & Latasa, 2015*). Indeed, both zooplankton and microbes are widely recognized for their essential role in biogeochemical processes that control flows of carbon, nitrogen and phosphorus (*Hébert, Beisner & Maranger, 2016b*) in wetland systems (*Schimel & Schaeffer, 2012*; *Herbert et al., 2015*). Since salinity is recognized as a primary determinate of both zooplankton (*Bate et al., 2002*; *Kimmel, 2011*; *Breckenridge et al., 2015*) and bacterial communities, salinization of wetlands might be expected to have particularly strong affects on wetland systems.

Despite the likely widespread dispersal of most microorganisms, a large review of fresh and marine species found little overlap between habitats, confirming that salinity acts as a large abiotic barrier for most microorganisms (*Logares et al., 2009*). Microbial functional groups also change along a salinity gradient (*Dupont et al., 2014*; *Eiler et al., 2014*; *Coci et al., 2005*; *Langenheder et al., 2003*) which suggests that increases in salinity in freshwater ponds could shift the abundance, richness and functional processes of bacterial communities that are critical in all ecosystems. However, the potential effects of changes in salinity on important downstream ecosystem functions, such as litter decomposition and carbon mineralization, are not well understood.

Rates of decomposition may differ as a function of salinity, the type of litter, micro- and macro-fauna present in the community, and the time since decomposition began. For instance, the home field advantage hypothesis (*Hunt et al., 1988*; *Gholz et al., 2000*) suggests that decomposition rate is most efficient when leaf litter is being decomposed in its natural habitat. That is, terrestrial species (e.g., *Acer sp.*) will decompose best in freshwater, while marine species (e.g *Fucus sp.*) will decompose fastest in marine systems. However, evidence for this hypothesis is quite mixed (*Franzitta et al., 2015*; *Lettice, Jansen & Chapman, 2011*; *Quintino et al., 2009*; *Reice & Herbst, 1982*; *Lopes et al., 2011*; *Connolly, Sobczak & Findlay, 2014*) and decomposition may be determined better by nitrogen and lignin content rather than salinity (*Stagg et al., 2018*).

Carbon mineralization also differs across wetland habitat type. Estuarine wetlands rapidly sequester carbon, accounting for approximately 30% of carbon sequestration in the lower USA (*Bridgham et al., 2006*), and they retain this stored carbon for longer
than other ecosystems (*Mcleod et al., 2011*). Although precisely calculating the carbon budgets of wetlands is complicated by their concomitant release of methane gas, they are nevertheless generally considered to serve as an important net carbon sink in the long term (*Mitsch et al., 2013*). Unfortunately, coastal and estuarine wetlands are vulnerable to biogeochemical changes due to SLR and other environmental perturbations and are rapidly being lost (*Hopkinson, Cai & Hu, 2012*). In addition, higher salinity soils often have lower levels of carbon mineralization and methane gas release (*Setia et al., 2011*; *Weston, Dixon & Joye, 2006*; *Al-Busaidi, Buerkert & Joergensen, 2014*; *Poffenbarger, Needelman & Megonigal, 2011*), although these results are not universal (*Chambers, Reddy & Osborne, 2011*). Regardless, understanding how carbon budgets may change as wetlands change is critical for understanding and mitigating impacts of climate change.

Our study examines the impacts of salinization on species diversity, community structure and associated ecosystem functions in coastal shallow freshwater wetlands. We examined how overwash events along with mixing of freshwater and saltwater taxa affect the diversity and composition of bacteria and zooplankton communities and downstream ecosystem functions. To test the effects of salinization on diversity and ecosystem function we performed a semi-natural mesocosm experiment in which we simulated wetlands with different salinities. We simulated the effects of salt-water incursions and the mixing of salt and freshwater communities by imposing two treatments: one that included a sample of both fresh and 13 psu plankton and microbes, and a second that was a sample of salt-only plankton and microbe communities. We quantified changes in zooplankton and bacteria communities and measured two representative ecosystem functions: carbon mineralization and litter decomposition. We expected that differences in species identities and diversity among patches would translate into differences in aggregate ecosystem functions (*Staddon et al., 2010*; *Symons & Arnott, 2013*; *Dodson, 1992*). To gain more clarity on how decomposition changes across salinities we tested the home field advantage hypothesis by measuring the decomposition of three species with different natural habitats over 6 weeks along a salinity gradient. Additionally, we hypothesize that differences in decomposition will be correlated with the associated microbial and zooplankton communities. Finally, to further enhance our understanding of how SLR and seawater overwash might affect the carbon cycle in the face of ongoing impacts from climate change, we examine how the zooplankton and bacterial communities correlate with carbon mineralization across the salinity gradient.

## METHODS

### Experimental set-up

Our experiment took place in North Carolina, USA. North Carolina is a suitable place for studying the effects of salinity because SLR is occurring faster there than in other regions on the US Atlantic coast (*Kemp et al., 2009*; *Kopp et al., 2015*).

We created 39 experimental ponds using 567 L stock watering tanks. Tanks were filled with 378 L of water from a hose; we recognize that by not sterilizing the water it is possible that bacteria were introduced in such a way that bacterial richness was disproportionately
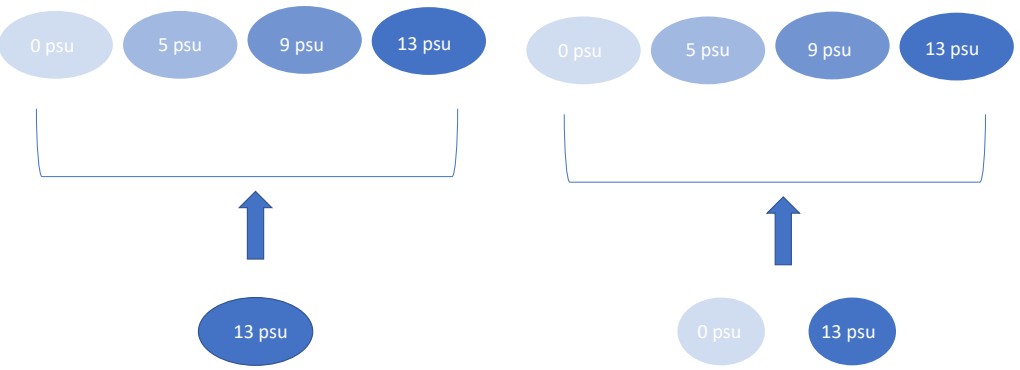

**Figure 1** **Experimental design showing the four salinity treatments and the two dispersal treatments.**
Arrows indicate mixing treatment. This experimental design was replicated four times, except for 5 psu
with mixed dispersal which was replicated three times.

increased in freshwater communities. Instant Ocean sea salt was used to generate salinity
treatments that closely matched the salinity of local coastal ponds (0, 5, 9 and 13 psu)
(*Albecker & McCoy, 2019*). Tanks were randomly assigned to receive one of the four salinity
treatments (0, 5, 9 and 13 psu), and each tank was initially seeded with zooplankton and
bacteria from a natural pond with matching salinity (e.g., at 5 psu treatment was seeded
with a community from a natural pond at 5 psu) located along the inner and outer banks
of North Carolina on May 3, 2015 (Table S1). (N.B. samples from two different ponds
were mixed for the highest salinity treatment).

We maintained "source" experimental ponds at 0 and 13 psu that were used to provide
the colonists for the other experimental ponds. These species mixing treatments consisted
of a "salt-only" plankton community which only received water from the 13 psu source
tanks or "mixed" plankton treatment which received an aliquot of water and plankton
consisting of equal volumes (each 50% of the total aliquot) sampled from the zero and 13
psu source tanks (Fig. 1). Species mixing treatments were applied every nine days for a total
of five species introductions over the course of the experiment. Plankton communities in
all experimental ponds were sampled prior to each new introduction event. We chose this
mixing regime to mimic the effects of saltwater over-wash and intrusion on freshwater
wetlands since salinization events may be common in coastal ponds (*Albecker & McCoy,*
*2019*) and likely represent the unidirectional movement of saltwater species into freshwater
communities. Each treatment combination was replicated four times, except for the
5 psu/mixed mixing treatment which only had three replicates due to a leak in one
experimental mesocosm.

To collect our initial zooplankton and bacteria from coastal ponds, we sampled along a
single 100 m transect at each pond taking twenty 1 L samples of water from within a foot of
the surface (most ponds were less than 2 ft deep at the time of sampling). We strained each
sample them through a 62.5 μm mesh filter. If a pond was too small to complete a full 100
m transect, a second transect was used. These samples served as the starting communities
for the experiment. In addition to samples from coastal ponds, the experimental tanks

**Table 1  Zooplankton abundance (mean standard deviation) per liter for each dispersal source tank (13 psu or 0 psu).** No mixing treatment was exclusively freshwater, instead a combination of half freshwater and half 13 psu.

| Source | 1st dispersal | SD | 2nd dispersal | SD | 3rd dispersal | SD | 4th dispersal | SD | 5th dispersal | SD |
|---|---|---|---|---|---|---|---|---|---|---|
| 13 | 1.2 | 1.7 | 2.35 | 2.5 | 1.8 | 3.3 | 1.1 | 1.5 | 1.6 | 2.2 |
| 0 | 3.4 | 7.1 | 7.24 | 9.9 | 4.1 | 6.1 | 11 | 18.8 | 4.6 | 6.9 |

were seeded with peat moss to provide a nutrient pulse and the tank bottoms were covered with sand as a benthic substrate. Mesocosms were covered with 60% shade cloth to prevent macroinvertebrates and other higher trophic level organisms from colonizing.

Species mixing consisted of a 2 L aliquot of water from the source tanks; due to natural dynamics in these tanks the actual abundances varied for each mixing event (Table 1). On June 1, 2015, prior to beginning the experiment, we detected very low zooplankton abundance from the first seeding in the 13 psu tanks, so we re-seeded with a new wild sample of zooplankton. To allow populations to stabilize, the experiment began 6 weeks after initial seeding. For 45 days, we sampled all experimental ponds every 9 days. We had a 9 day sampling regime because this is long enough for most zooplankton species to complete at least one-generation cycle (*Thompson & Shurin, 2012*). Prior to sampling, we mixed each tank by stirring them in a circular motion around the perimeter five times. Twenty liters (approximately 5% of total volume) of water was sampled from the water column at 20 random locations using an integrated tube sampler. After mixing we sampled from the center of the water column; we don't expect our tanks to be stratified due to their depth (<0.6 m) (*Snucins & John, 2000*). Samples were condensed through a 62.5 µm filter into 25 mL containers. Zooplankton from each tank at the time of sampling were preserved in 10% formalin.

Zooplankton were counted in three five mL subsamples and identified to the lowest taxonomic level possible (order, family, or genus when feasible using *Johnson & Allen (2012)* and *Pennak (1953)*); however, for all analyses either family or order were used. Based on some known functional redundancy within zooplankton orders and family level taxonomic groupings (e.g., *Barnett, Finlay & Beisner, 2007*) we expected this level of resolution to be sufficient to capture major impacts of changes in assemblages on ecosystem functions.

## Bacterial sampling

Bacterial sampling was concurrent to zooplankton sampling. At each sampling event we collected 1 L of water from each tank by scooping a bottle several times in the tank until we had 1 L. Each 1 L bottle of water was homogenized and 200 mL of the water sample was concentrated onto 0.22 µm filters within 24 h of field sampling (Supor-200; Pall Gelman, East Hills, NY). Filters were transferred into two mL sterile tubes and stored at −80 °C until molecular analyses was completed.

### Bacterial community sequencing

To examine shifts in bacterial community composition and diversity, bacteria in each mesocosm were characterized using paired-end targeted Illumina sequencing of the 16S

rRNA gene (bacteria, archaea) (*Caporaso et al., 2011*). We extracted DNA from filters collected at 3 of the 6 time points representing the initial, middle, and final sampling days (Days 0, 18, 45). We extracted and purified the DNA from 0.22 μm supor filters from each mesocosm using the PowerWater DNA Isolation Kit (MO BIO Laboratories, Inc CA). We used this DNA as a template in PCR reactions. To characterize particle and free-living organism communities, we used barcoded primers (515FB/806RB) originally developed by the Earth Microbiome Project (*Caporaso et al., 2012*) to target the V4-V5 region of the bacterial 16S subunit of the ribosomal RNA gene (*Apprill et al., 2015*; *Parada, Needham & Fuhrman, 2016*). This primer set targets Bacteria and Archaea. For this study, we focused on the bacteria. PCR products were combined in equimolar concentrations and sequenced using paired-end (2 ×250 bp) approach using the Illumina MiSeq platform at the Indiana University Center for Genomics and Bioinformatics.

Raw sequences were processed using the Mothur pipeline (version 1.39.4 *Kozich et al., 2013*; *Schloss et al., 2009*). Contigs from the paired end reads were assembled and quality trimmed using an average quality score, sequences were aligned to the Silva Database (version 123) (*Quast et al., 2012*), and chimeric sequences were removed using the VSEARCH algorithm (*Rognes et al., 2016*). Next, we created operational taxonomic units (OTUs) by splitting sequences based on taxonomic class and then clustering these OTUs by 97% sequence similarity. To estimate observed bacterial richness, we rarefied abundances to the minimum sequence depth of 13,000 reads. The original sequence data set had 12 million total sequences with 95,000 sequences per sample on average. After initial filtering there were 8.1 million sequences with 58,000 sequences on average per sample.

## Statistical analyses
### Alpha diversity
We used richness to explore alpha diversity. Zooplankton taxonomic order richness was evaluated using a generalized linear model with a quasi-Poisson error distribution; a quasi-Poisson distribution was used because data were under-dispersed. For all Poisson distributed models, we evaluated under/over dispersion of our error distribution by looking at the ratio of Pearson's residuals and the residual degrees of freedom (*Bolker, 2008*). We defined observed bacterial richness by the number of different OTUs in a community. Over-dispersed observed bacterial richness was modeled using a negative binomial error distribution. Analyses were conducted using the lme4 (*Bates et al., 2015*) and MASS (*Venables & Ripley, 2013*) packages, respectively, in the R statistical programming environment (*R Core Team, 2016*). Richness was modeled as a function of salinity, mixing treatment, time, and interactions between time and salinity and salinity and mixing. We included a random effect of replicate over time which allows the intercept and slope of each replicate to vary; this takes into account the grouping of repeated measures within each tank. For analysis, parameter-specific *p*-values in a fully parameterized model were used to determine the significance of predictors. We include results for Shannon Diversity in the Section S9.3.3.

### Testing for effects on community composition

Community structure of both bacterial and zooplankton communities, including visualizing community turnover over time and turnover between treatments, was evaluated using Principle Coordinates Analysis (PCoA) with Bray–Curtis dissimilarity. The PCoA graphs (Figs. 2 and 3) are generated based on a single ordination. Variation explained by mixing, salinity, and time was analyzed using a permutational multivariate analysis of variance (PERMANOVA). These analyses were conducted in R using the Vegan 2.3.3 package (*Oksanen et al., 2016*). We used indicator species analysis to identify which bacterial taxa were most representative of each salinity treatment (*Dufrêne & Legendre, 1997*). We used the Labdsv package in R to run the analysis (*Roberts, 2016*). For the indicator species analysis, we only included bacterial taxa with a relative abundance greater than 0.05 when summed across all tanks.

## Ecosystem function

We assessed the effects of salinity, zooplankton, bacteria, and species mixing on ecosystem functions using two different proxies for ecosystem function: decomposition amount and carbon mineralization of the final communities.

### Decomposition

Leaf litter from three plant species were used in each tank to represent different habitat types: *Spartina alterniflora* found in salt marshes, *Acer rubrum* found in freshwater wetlands, and *Phragmites australis* found in both fresh and saltwater wetlands. We wanted to represent the three natural habitats along our gradient to understand the potential for differential effects of mixing on ecosystems along this salinity gradient. Leaves were harvested and air-dried in late May, 2015. Each tank received standardized amounts of leaf litter (*Acer rubrum*: 4.00 g; stdev ±0.01; *Spartina alterniflora*: 6.99 g stdev ±0.03; *Phragmites australis*: 10.01 g stdev ±0.03). *Phragmites australis* and *Acer rubrum* were housed in 24 inch mesh mariculture bags, while *Spartina alterniflora* was housed in windowscreen bags with smaller holes since *Spartina alterniflora* was not securely retained within the mesh mariculture bags. Leaf litter remained in the tanks for the duration of the experiment. On day 45, the bags were removed, air-dried, oven dried for 48 h, and then weighed. Decomposition was quantified as the proportion of leaf dry weight loss (housed in decomposition bags) from the beginning to end of the experiment.

To determine the relationship between proportional change in dry weight and the predictor variables: observed bacterial richness, zooplankton richness, salinity, mixing treatment, leaf litter type, and the interaction of salinity and leaf litter type, we used a beta regression with the package betareg (*Grün, Kosmidis & Zeileis, 2012*) (because the response is continuous and bounded between 0 to 1). We included the interaction between salinity and leaf litter type because we expected leaf litter would decompose differently in its native vs non-native abiotic conditions (e.g., *Acer rubrum* in freshwater verses the 13 psu water).

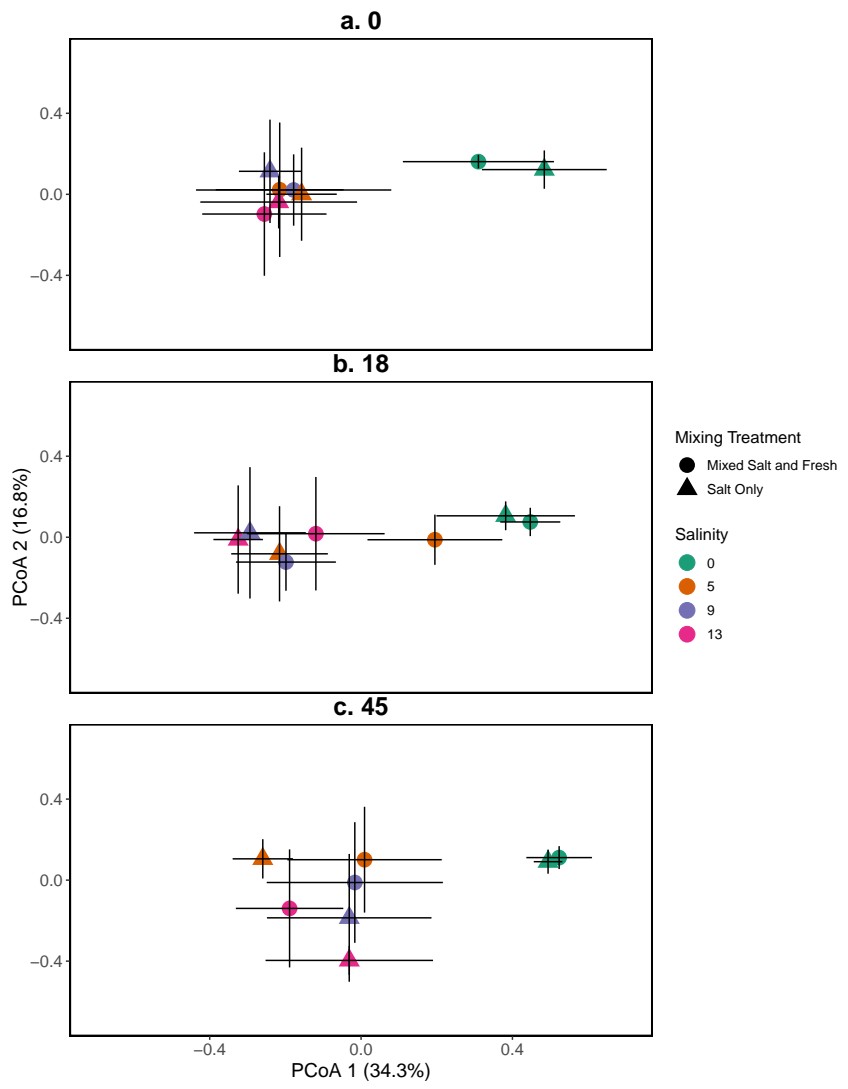

**Figure 2  PCoA for the relationship between zooplankton communities and salinity at three time points.** Zooplankton communities are represented by their centroid. Error bars show standard deviation. (A–C) represent different sampling days: (A) day 1 (starting community structure), (B) day 18, and (C) the final day (day 45). Shapes indicate dispersal treatment: circles show mixed salt and freshwater, triangles show salt water only mixing. Colors represent salinity treatment. Axes are PCoA 1 ($x$-axis) and PCoA 2 ($y$-axis).

### Carbon mineralization

On the final sampling day (day 45), we measured the amount of $CO_2$ respired from the aquatic communities using a laboratory-based bottle assay. Wheaton bottles (125 mL) fitted with septa were filled with water samples (25 mL) from each mesocosm tank. The $CO_2$ concentration readings were determined using an LI-7000 Infrared Gas Analyzer (IRGA). On the day of collection (the final day of the experiment), bottles were filled with 25 mL of mesocosm tank water, and the gas samples were collected and analyzed immediately using the IRGA to determine the baseline $CO_2$ concentration. A syringe was

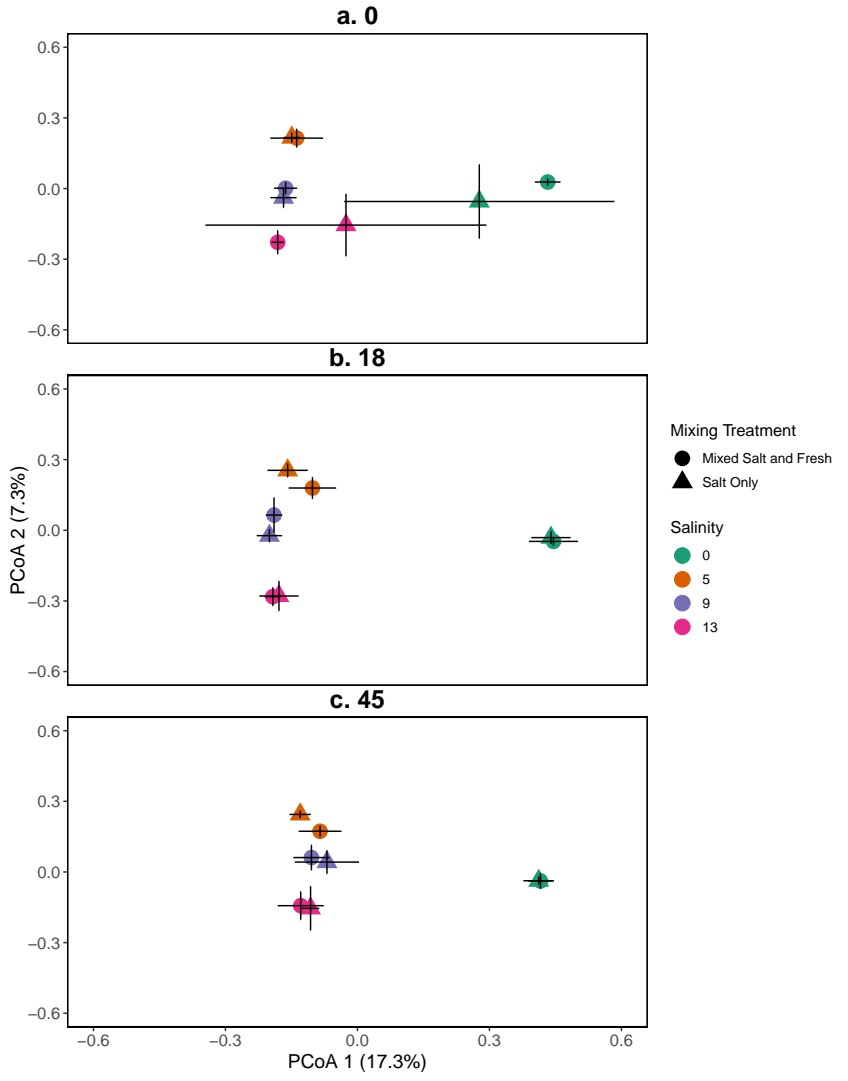

**Figure 3** **PCoA for the relationship between microbial communities and salinity at three time points.**
Points represent the centroid of the bacterial community structure. Error bars represent standard devia-
tion. Panels show different sampling days: (A) day 1 (starting community structure), (B) day 18, and (C)
the final day (day 45). All shapes and colors follow Fig. 2.

inserted into the septa and the headspace gas was mixed 3 times before pulling a sample and
beginning analysis using the IRGA. This process was repeated on days 1, 3, and 7 following
collection in order to determine $CO_2$ respiration rates over time. To determine the $CO_2$
production of each aquatic sample, the initial reading was subtracted from the analyzed
day's reading. We made a calibration curve with a known concentration of $CO_2$ over a set
of known volumes to get the calibration curve. Then, the unknown gas samples from our
sample set was compared to the known sample. To calculate the $CO_2$ respiration rate, the
concentration of $CO_2$ calculated from the calibration curve was converted to volume units

(ppm) using the following equation:

$$Cm\left(CO_2^{-C}L_{headspace}^{-1}\right) = \frac{Cv \cdot M \cdot P}{R \cdot T}$$

where Cm is carbon mineralization, Cv is the volume (ppm) of $CO_2$, M is the molecular weight of carbon, P is 1 atm, R is the universal gas constant (0.0820575 L atm K mole), and T is the incubation temperature in Kelvin. This value is then multiplied by the volume of the incubation chamber (L) and divided by the weight of water in the bottle used in the incubation to get $\mu$g $CO_2$-C gram$^{-1}$ water. To get the rate, this number is divided by the number of days incubated to get $\mu$g $CO_2$-C gram water$^{-1}$ day$^{-1}$.

We ran a linear model for carbon mineralization with zooplankton richness, microbial richness, mixing treatments, and salinity as predictors. In order to meet the assumptions of normality we log transformed the carbon mineralization data. There was a single replicate of a 9 psu tank that received the salt-only mixing treatment that was removed from the carbon mineralization analysis due to a missing data point.

After seeing the data we ran an *a posteriori* exploratory analysis where we used the same model as above but included a squared (quadratic) term for salinity to examine evidence of an intermediate minimum. We used AIC to compare models with and without the quadratic term.

## RESULTS

### Alpha diversity
#### *Zooplankton community*
Differences in zooplankton family richness was not well described by any of the predictors used in our analyses (all $p > 0.05$, Fig. 4); for model parameter estimates see Table S2. We find similar results using Shannon Diversity (see Section S9.3.3) For source tank richness see Fig. S1.

#### *Bacterial community*
Observed species richness for the bacterial community increased as salinity increased (estimate (log scale) = 0.035, standard error (log scale) = 0.008, $z = 4.0$, $p = 4.97e - 05$), and over time (estimate (log scale) = 0.008, standard error (log scale) = 0.002, $z = 4.07$, $p = 4.51e - 05$) (Fig. 5). However, the observed increase in richness over salinity reversed by the end of the experiment (Salinity:time: estimate (log scale) = $-0.001$, standard error (log scale) = 0.0003, $z = -4.2$, p=2.33e $- 05$) (Fig. 5C). There were no clear differences as a result of the mixing treatments nor the interaction between salinity and mixing treatment ($p > 0.05$, see Table S2 for coefficients). For source tanks richness see Fig. S2. We find similar results when using Shannon Diversity (see Section S9.3.3).

### Community composition
#### *Zooplankton community*
Zooplankton communities initially aggregated into two distinct groups: a freshwater group and a group consisting of all other salinities (Fig. 2). However, by the final day, the low salinity (5 psu) ponds receiving the mixed species treatment were more similar in

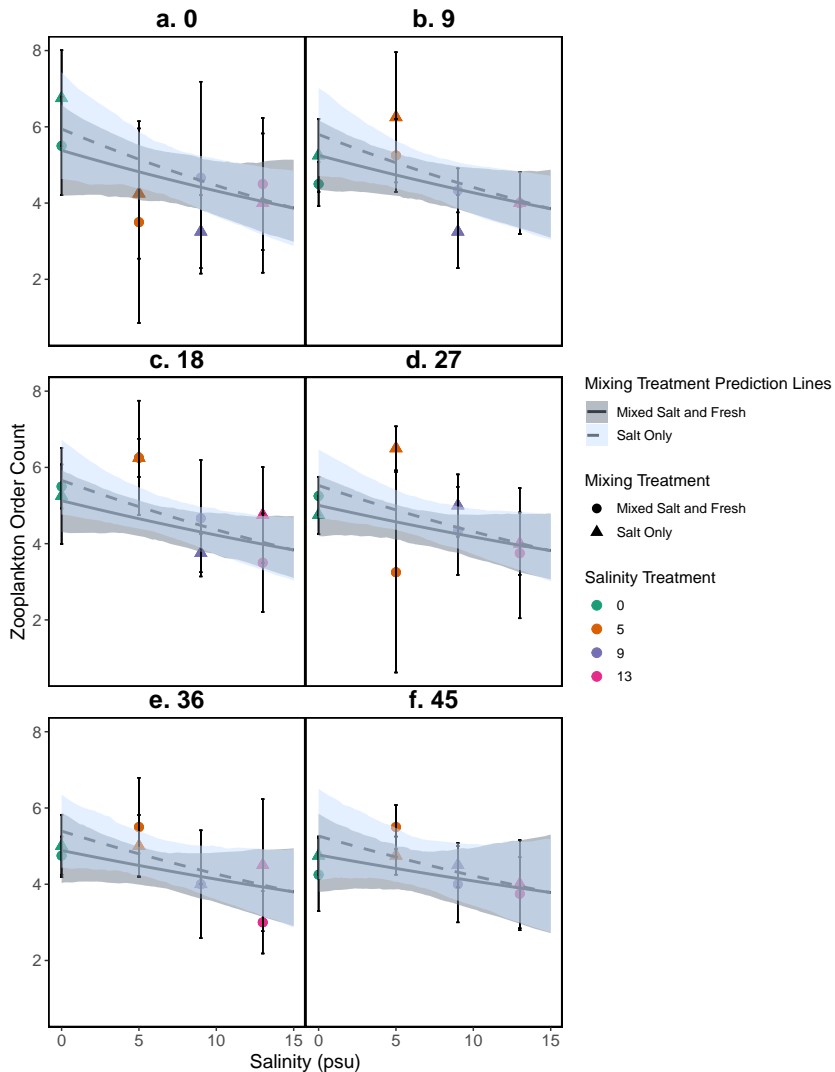

**Figure 4  Relationship between zooplankton order count and salinity.** (A–F) represent zooplankton richness (mean ± standard deviation) at a single sampling day. Color indicates the salinity treatment. Shape indicates mixing treatment: circles show salt and fresh water community mixing and triangles show salt-only mixing. Lines are model estimates: solid lines represent predictions for the mixed fresh and salt water treatment and dotted lines show predictions for the salt-only mixing treatment. Predicted lines are transformed back to original scale. Envelopes show bootstrap 95% confidence intervals.

composition to the freshwater community. The 9 and 13 psu salinity treatments remained distinct from freshwater treatments with regards to their community structure. PCoA one explained 31% of variation and PCoA two explained 14%. PERMANOVA results suggest that salinity contributed most to variation in zooplankton communities ($R^2 = 0.23$, $p < 0.0001$). In contrast, the effects of the mixing treatment ($R^2 = 0.03$, $p < 0.0001$), time ($R^2 = 0.029$, $p < 0.0001$), and the interaction between time and salinity ($R^2 = 0.019$, $p < 0.0001$) on community variance were relatively more modest. While we observe an effect of the two and three way interactions between salinity, mixing, and time (all $p < 0.05$, except

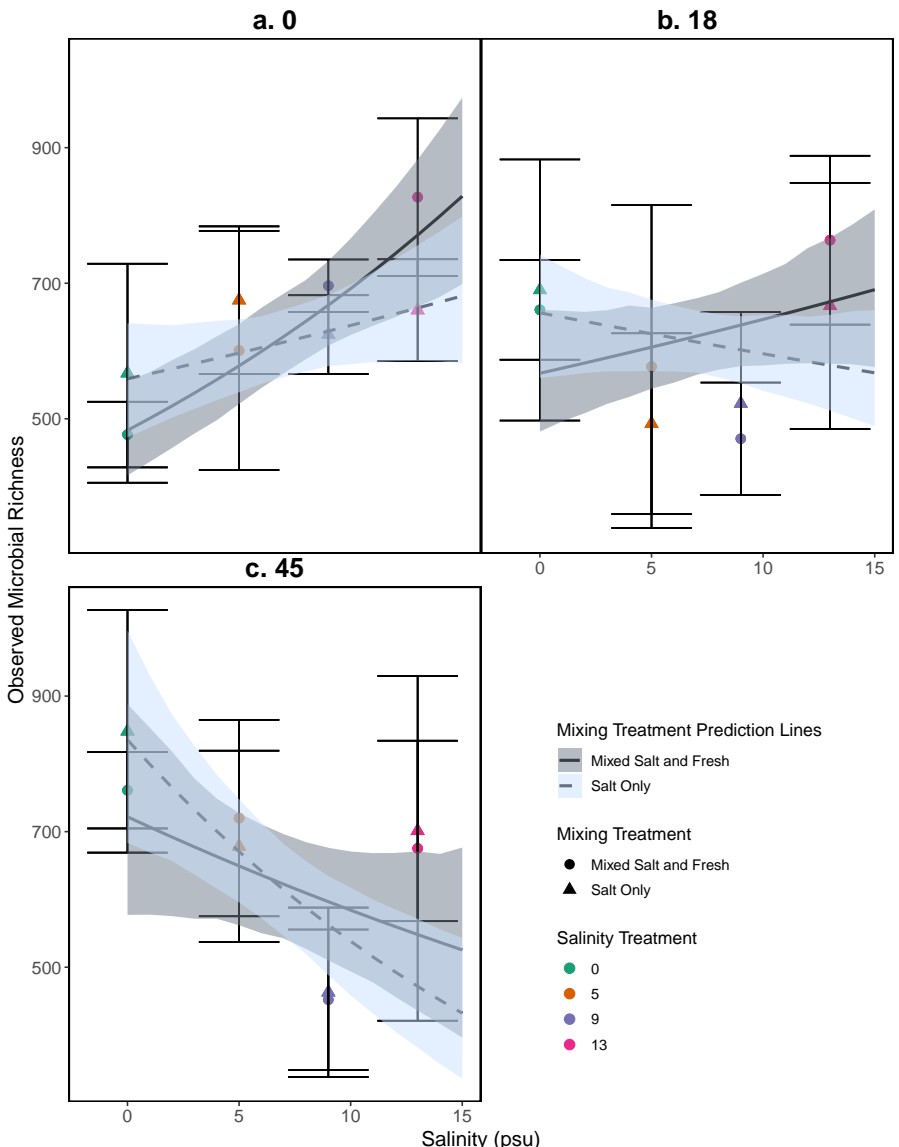

**Figure 5** **Relationship between observed microbial richness and salinity.** (A–C) represent different sampling days: (A) day 1 (starting community structure), (B) day 18, and (C) the final day (day 45). Data and model estimates are shown on the original count scale. All symbols and colors match those in Fig. 4.

the interaction of dispersal and salinity $p > 0.05$), the total amount of variation explained is quite small ($R^2 < 0.01$ in all cases). For source tanks alone and source tanks in relation to all other tanks see Figs. S3, S4.

### Bacterial community

A mantel test revealed that zooplankton and bacterial communities were positively correlated (mantel test: $r = 0.409$, $p = 0.001$). For the bacterial community the main effects of salinity and time account for the most variation (PERMANOVA, salinity: $R^2 = 0.115$, $p = 0.001$, time: $R^2 = 0.052$, $p < 0.001$). Different mixing treatments did not

have a clear differential effect on bacterial community structure (PERMANOVA, mixing: $R^2 = 0.007$, $p = 0.786$). The bacterial communities in the treatment tanks separated into salt vs. freshwater environments along the primary axis, which explained 17.3% of the variation among communities (Fig. 3). Distinct bacterial communities grouped according to increasing salinity (5, 9, 13 psu) and separated along the secondary axis, which explained 7.3% of the variation in bacterial community composition. For information on the source tanks see the Figs. S5 and S6.

Indicator species analysis identified 225 bacterial taxa (OTUs) that were representative of salinity treatment (Table S3). Associating these organisms with a salinity level can identify key taxa contributing to shifts in bacterial community structure. Due to the great diversity of bacterial communities, many bacterial sequences were unresolved to the 'species' level (operationally defined at 97% sequence similarity) but instead were classified according to the closest known sequence match. Proteobacteria (phylum) was the strongest indicator of zero salinity (IndVal = 0.991). Rhodospirillales (class) was the second highest indicator taxon (IndVal = 0.990) and *Polynucleobacter* (genus) was the third highest indicator (IndVal = 0.983) of the zero salinity treatment. Unclassified Betaproteobacteria (class; IndVal = 0.936) represented the salinity 5 environments, followed by *Flavobacterium* (genus; IndVal = 0.889) and Alcaligenaceae (family; IndVal = 0.852). Bacteria representing Salinity 9 and 13 environments were less clear. In the more saline treatments, 5 of 8 OTUs were unclassified and were unresolved beyond the Bacterial domain (Table S3). Planctomycetes had the third highest indicator value in the 9 psu treatments, and was only 1 of 4 classified OTUs indicative of that treatment (phylum; IndVal = 0.804). The presence of this phylum in 9 psu tanks represents a slight shift in community dominance from fresh to salt-tolerant taxa; however, the other top 3 indicator taxa of salinity 9 tanks were unclassified, so conclusions regarding key bacterial taxa involved remain elusive. Salinity 13 also had unclassified taxa identified in the top five indicators species; there were 2 classified and 2 unclassified taxa. The 2 classified taxa were Haliea (genus; IndVal = 0.869) and Alphaproteobacteria (class; IndVal = 0.928). Genus Haliea is a Gammaproteobacteria (class) with species isolated from aquatic marine environments.

## Ecosystem function
### Decomposition
As bacterial richness increased the proportion of leaf mass remaining decreased, representing an increase in decomposition (estimate (log-odds scale) = -0.0007, standard error (log-odds scale) = 0.0002, $z = -3.04$, $p = 0.002$). As salinity increased, mass change decreased (estimate (log-odds scale) = 0.043, standard error (log-odds scales) = 0.018, $z = 2.38$, $p = 0.017$). The salt-only mixing treatment had lower overall decomposition (less mass lost) than the mixed mixing treatment (estimate (log-odds scale) = $-0.19$, standard error(log-odds scale) = 0.086, $z = -2.26$, $p = 0.02$). *Spartina alterniflora* lost less material than *Acer rubrum* leaves (estimate:log link 1.1, standard error:log link 0.18, $z = 5.9$, $p << .001$) (Fig. 6). In contrast, we were unable to detect an affect of zooplankton
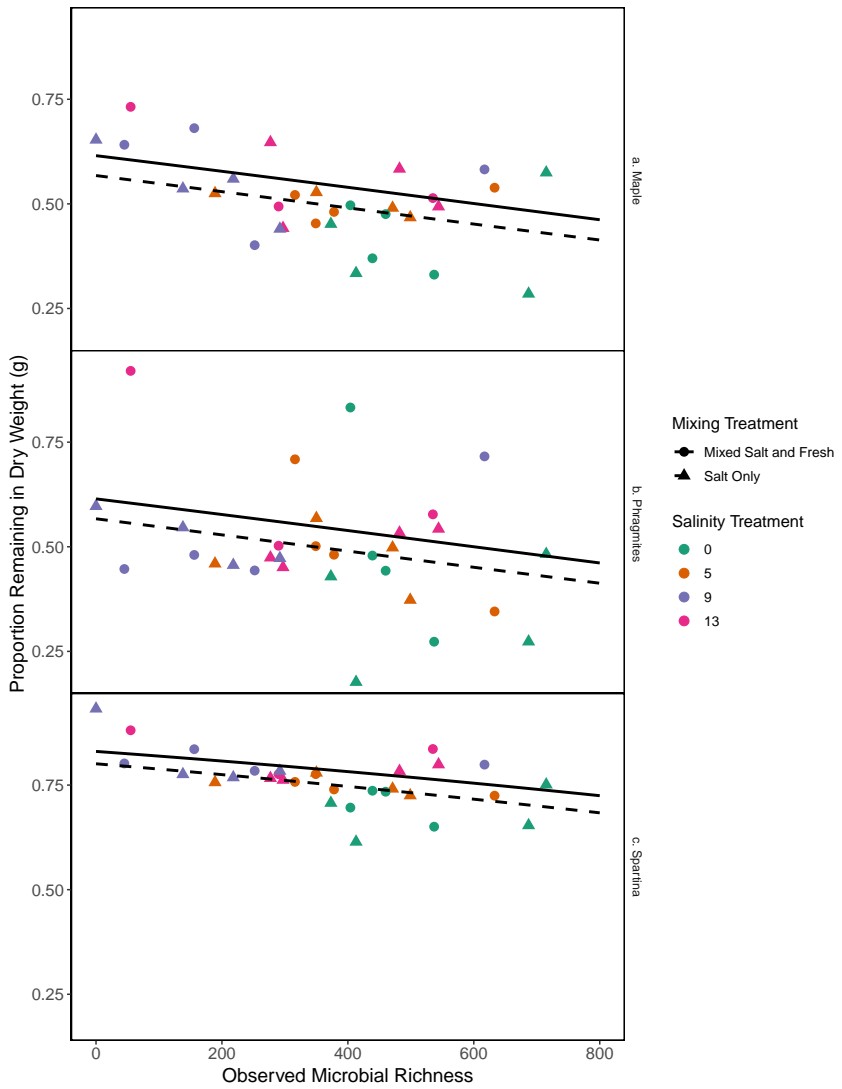

**Figure 6  Proportion of leaf litter remaining in relation to microbial richness.** The $y$-axis shows the proportion of leaf litter remaining at the end of the experiment; the more leaf litter remaining the less decomposition occurred. Panels represent change in weight in each leaf litter type: (A) *Acer rubrum*, (B) *Phragmites australis*, and (C) *Spartina alternaflora*. Points are colored by salinity treatment and shaped by leaf litter type. Lines represent model predictions: solid lines represent predictions for the mixed fresh and salt water treatment and dotted lines show predictions for the salt-only mixing treatment. The estimates shown here were obtained using average zooplankton richness (4.5) and mean salinity (6).

richness or any of the interaction terms with leaf type (all $p > 0.05$). Overall the model accounted for a large fraction of the variation (pseudo $R^2 = 0.66$).

### *Carbon mineralization*
In our first *a priori* model we found that carbon mineralization increased with observed bacterial richness (estimate: 0.003, standard error: 0.001, $t = 2.78$, $p = 0.008$) (Fig. 7). Overall model fit was moderate (adjusted $R^2 = 0.31$, $F - statistic = 4.4$ on 5 and 32 DF).

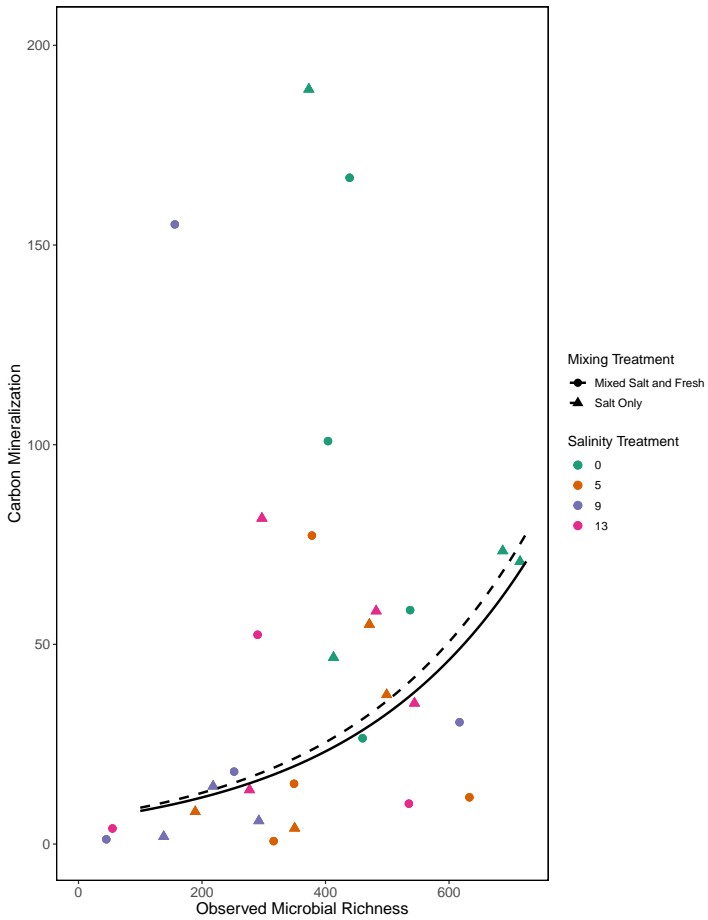

**Figure 7 Carbon mineralization given observed microbial richness.** Points are colored by salinity treatment. Lines represent model predictions: solid lines represent predictions for the mixed fresh and salt water treatment and dotted lines show predictions for the salt-only mixing treatment. The estimates shown here were obtained using average zooplankton richness (4.5) and mean salinity (6).

We were unable to detect an effect of zooplankton richness, mixing treatment, or salinity on carbon mineralization (all $p > 0.5$).

However, in our exploratory model we found that carbon mineralization decreased in the mid-salinity treatments (Fig. 8) (salinity$^2$: estimate: 8.2, standard error:1.4, $t = 5.9, p << 0.001$) and that carbon mineralization increased with zooplankton richness (estimate:0.5, standard error:0.16, $t = 3.1, p = 0.003$). This model explained more variation than our *a priori* model (adjusted $R^2 = 0.4$, $F - statistic = 14.4$ on 5 and 84 DF). We were unable to detect an effect of microbial richness, mixing treatment, or the main affect of salinity on carbon mineralization (all $p > 0.5$). Based on AIC, the second model with the squared salinity term, has more support (Delta AIC = 30).

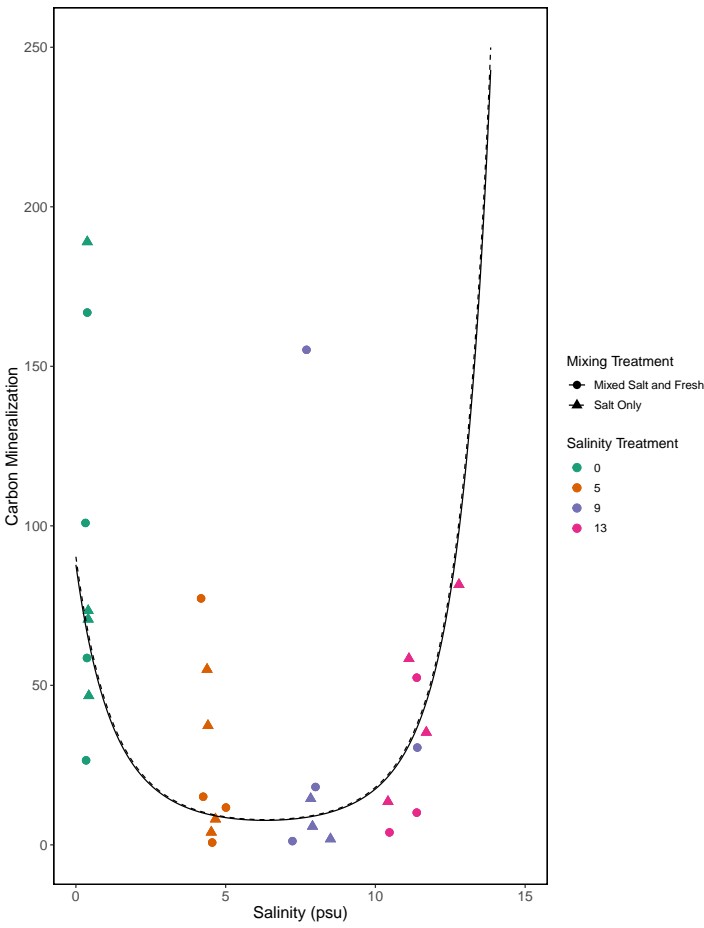

**Figure 8** **Exploratory analysis of the relationship between carbon mineralization and salinity.** Points are colored by salinity treatments and shapes by mixing treatment. The estimates shown here were obtained using average zooplankton richness (4.5) and mean observed bacterial richness (380.4).

## DISCUSSION

Understanding how extreme environmental gradients and changing patterns of connectivity can influence community structure and ecosystem functions is becoming increasingly important as species assemblages shift to keep pace with climate change (*Root et al., 2015*). While the mixing of previously distinct communities from environmental change may have dire consequences for some species (*Cahill et al., 2012*), an increased capacity to maintain ecosystem functions in the face of those same environmental perturbations may also be expected due to introduction of redundant or tolerant species (e.g. *Thompson & Shurin, 2012*; *De Boer et al., 2014*; *Mansour et al., 2018*).

Our results for zooplankton diversity and observed microbial richness patterns are consistent with communities that are determined by strong abiotic filters (Figs. 5 and 4) (*Leibold, Chase & Ernest, 2017*). Indeed, we found a clear delineation between freshwater and brackish water in our experiment (Figs. 2 and 3) which suggests that abiotic filters are

a strong and critical regulating force of the composition of zoo- and bacterio-planktonic communities at the fresh-brackish water interface. While we expected an increase in species richness in low to mid salinity pools due to sampling from a more diverse species pool (mixed salinity), the effect of species mixing in this study was likely masked by the strong effect of salinity on community composition (*Mouquet & Loreau, 2003*). Additionally, our experimental protocol permitted salinities and biotic communities to stabilize, which may have further buffered experimental pools against invasion (Fig. S7). Although a larger regional species pool (fresh and salt water species) might be expected to positively influence local diversity and function, fresh or salt water systems that have low levels of disturbance might be further resistant to invasion by new taxa (*Symons & Arnott, 2013*; *Symons & Arnott, 2014*) because of strong priority effects and competitive dominance hierarchies (e.g. *Geange & Stier, 2009*). Interestingly, we only observed changes in community structure in the 5 psu zooplankton community. Specifically, this community became more similar to a freshwater community in the mixed-salinity mixing treatment (Fig. 2). In contrast, the 13 psu and 0 psu salinity communities did not change over time, suggesting that new species are unable to easily colonize and establish in these highly filtered and stable environments.

Different microbial taxa were representative of each of the four different salinity levels, supporting previous work that suggests salinity tolerance is a specialized trait that determines bacterial community composition (*Martiny et al., 2015*). In the freshwater treatment one of the key indicator taxa, the Proteobacteria phylum, is the most diverse phylum of bacteria both in terms of taxonomic and functional diversity. Within the phylum Proteobacteria, we found Rhodospirillales, which includes many species that contain photosynthetic pigments and function as photoheterotrophs. Alternatively, the main indicator in the 5 salinity treatment, Betaproteobacteria class, consists of aerobic or facultative bacteria, which are capable of living in dynamic (redox) environments. The taxa found in salinity 5 are not characterized as existing in any one specific salinity. This may be attributed to the bacteria in the salinity 5 tanks being able to persist through the salinity change from fresh to salinity 5. For both 9 and 13 salinity we were unable to resolve the taxa of the most abundant OTUs. This suggests that less is known about these habitats in general and perhaps mid-salinity estuaries require more studies.

While it is not surprising that abiotic filtering had strong effects on community structure in our study, this study expands our understanding about how coastal systems may be affected by changes in salinity and species mixing. The observed changes in richness across salinity, in part, led to changes in ecosystem function. Indeed, in contrast to the responses of zooplankton, we found that bacterial richness increased with salinity, and that this increase in species richness was correlated with amount of decomposition. This result lends support to the hypothesis that changes in biodiversity can affect ecosystem function (*Mouquet & Loreau, 2003*). This effect is even more interesting because it acts inversely to the effect of salinity; as salinity increased, decomposition decreased overall (Fig. 6). That bacterial richness increased with increased salinity and that decomposition amount increased with increased bacterial richness in our system suggests there is some small compensation by bacteria that is mitigating the effect of salinity. However, the effect may be temporary because the increase in richness over salinity is reduced over time (Fig. 5). The smaller

difference in richness across salinities from the beginning to the end of the experiment (Fig. 5: Day 0 and 45) is driven by larger increases in richness in the freshwater treatments compared to the other treatments. However, because the freshwater communities did not become more similar to the salt communities over time (Fig. 3), it is unlikely that the increase in observed bacterial richness is due to mixing of species pools via the mixed treatments. Instead it is likely that rare taxa, which we didn't detect at the beginning, become dominant in intermediate salinities (*Rocca et al., 2019*) and that there was higher immigration from natural sources to freshwater treatments than other treatments. We do, in fact, expect passive dispersal via wind (*Nemergut et al., 2013*). Another line of evidence supporting the idea that influxes from high saline environments can change ecosystem function is that the salt-only mixing treatments had lower decomposition than the other mixing scenarios. Based on the home-field advantage hypothesis we expected differential leaf litter decomposition based on the leaf litter's native habitat (e.g., *Acer rubrum* in freshwater); however, we found no detectable differences in decomposition among different leaf litter types as a function of salinity. There is very mixed evidence for the home-field advantage hypothesis generally though, so it comes as no surprise that we also were unable to find conclusive results. Instead, the relationship between habitat and decomposition may be better described along a continuum of decomposer-litter interactions (*Freschet, Aerts & Cornelissen, 2012*) or by C:N and C:P ratios of the litter (*Kennedy & El-Sabaawi, 2017*).

Bacterial communities are known to be important in linking terrestrial, fresh and marine carbon cycles through transport, mineralization, and storage of carbon (*Ardón, Helton & Bernhardt, 2016*). Consistent with this expectation we found a positive correlation between bacterial communities and carbon mineralization in our *a priori* model. While zooplankton communities have also been directly linked to carbon mineralization (*Jonsson et al., 2001*) and carbon cycling (*Six & Maier-Reimer, 1996*), they may only account for a small proportion of total mineralization (*Jonsson et al., 2001*). In our first model we did not find a direct link between zooplankton richness and carbon mineralization; this is likely a consequence of small sample sizes and small expected direct effect of zooplankton on total carbon mineralization. However, in our exploratory model, when we considered a quadratic term, we were able to detect a positive relationship with zooplankton richness and carbon mineralization. We also saw a decrease in carbon mineralization at mid-salinity compared to either extreme in our exploratory model. This result leaves room for more specific experiments to determine if this is repeatable and what mechanisms could cause a unimodal response. This highlights the need for future work on biodiversity-ecosystem functions to both clarify mechanism and better quantify the importance of exploring multiple trophic levels.

## CONCLUSIONS

This study provides an important step toward understanding how mixing of communities along a salt gradient will affect local and regional patterns of diversity and ecosystem function. Future research should include perturbations such as variability in salinity within

a single season, perhaps explicitly testing predictions made over changing heterogeneous landscapes as presented by *Thompson & Gonzalez (2017)*. Additionally, our study further supports recent calls for experiments that explicitly use traits or taxonomic groups related to functions of interest to investigate links to ecosystem functions (e.g. *Violle et al., 2007*; *Hébert, Beisner & Maranger, 2016a*). Our results highlight the need to better understand how changes in the abiotic environment and mixing of novel communities interact to affect how ecosystems (such as coastal ponds) respond to the rapid environmental changes and accelerating rates of global change.

## ACKNOWLEDGEMENTS

We would like to thank Spencer Wilkinson, Amanda Dunn, and Mary-Grace Lee for help with data collection. We would like to thank the Coastal Studies Institute (CSI) for use of their space and Mike Piehler and Corey Adams for logistical help at CSI.

### Funding

This project was funded by a Sigma Xi grants in aid of research (G201503151193315) and the Lerner Gray Memorial fund of the American Museum of Natural History. Michael W. McCoy and Ariane L. Peralta both have funding through East Carolina University. The funders had no role in study design, data collection and analysis, decision to publish, or preparation of the manuscript.

### Grant Disclosures

The following grant information was disclosed by the authors:
Sigma Xi: G201503151193315.
American Museum of Natural History.
East Carolina University.

### Competing Interests

The authors declare there are no competing interests.

### Author Contributions

- Jo A. Werba conceived and designed the experiments, performed the experiments, analyzed the data, prepared figures and/or tables, authored or reviewed drafts of the paper, and approved the final draft.
- Alexandra L. Stucy performed the experiments, analyzed the data, authored or reviewed drafts of the paper, and approved the final draft.
- Ariane L. Peralta conceived and designed the experiments, authored or reviewed drafts of the paper, and approved the final draft.
- Michael W. McCoy conceived and designed the experiments, authored or reviewed drafts of the paper, and approved the final draft.

## Data Availability

Data and code are available in the following GitHub repositories:

https://github.com/PeraltaLab/CSI_Dispersal.

Go to the "Werba_Data_Analyses_Manuscript_Effects of diversity and coalescence of species assemblages" folder.

https://github.com/PeraltaLab/CSI_Dispersal/tree/master/Werba_Data_Analyses_Manuscript_Effects%20of%20diversity%20and%20coalescence%20of%20species%20assemblages.

## Supplemental Information

Supplemental information for this article can be found online at http://dx.doi.org/10.7717/peerj.8608#supplemental-information.

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
